# Novel Deep-Water Tidal Meter for Offshore Aquaculture Infrastructures

**DOI:** 10.3390/s22155513

**Published:** 2022-07-24

**Authors:** Javier Sosa, Juan-A. Montiel-Nelson

**Affiliations:** Institute for Applied Microelectronics (IUMA), University of Las Palmas de Gran Canaria, 35015 Las Palmas de Gran Canaria, Spain; montiel@iuma.ulpgc.es

**Keywords:** ocean tides, waves, underwater sensors, accelerometer, MEMS, offshore aquaculture infrastructures

## Abstract

This paper presents a tidal current meter that is based on the inertial acceleration principle for offshore infrastructures in deep water. Focusing on the marine installations of the aquaculture industry, we studied the forces of tides at a depth of 15 m by measuring the acceleration. In addition, we used a commercial MEMS triaxial accelerometer to record the acceleration values. A prototype of the tidal measurement unit was developed and tested at a real offshore aquaculture infrastructure in Gran Canaria, which is one of the Canary Islands in the Atlantic Ocean. The proposed tidal measurement unit was used as a recorder to assess the complexity of measuring the frequency of tidal currents in the short (10 min), medium (one day) and long term (one week). The acquired data were studied in detail, in both the time and frequency domains, to determine the frequency of the forces that were involved. Finally, the complexity of the frequency measurements from the captured data was analyzed in terms of sampling ratio and recording duration, from the point of view of using our proposed measurement unit as an ultra-low-power embedded system. The proposed device was tested for more than 180 days using a lithium-ion battery. This working period was three times greater than the best alternative in the literature because of the ultra-low-power design of the on-board embedded system. The measurement accuracy error was lower than 1% and the resolution was 0.01 cm/s for the 0.8 m/s velocity scale. This performance was similar to the best Doppler solution that was found in the literature.

## 1. Introduction

Since ancient times, humanity has considered the ocean to be a source of potential resources. In addition to the classic transport, communication and fishing applications, new applications such as power generation and aquaculture have grown in importance. Modeling the behavior of the oceans has been and continues to be a hot topic of research [1]. It is well known that the study of winds, waves, tides and currents determines the viability of ocean infrastructures, especially offshore aquaculture facilities. Once facilities have been installed, these variables must be monitored to prevent or warn of breakages.

Marine infrastructures, particularly offshore aquaculture facilities, require a high degree of knowledge about the forces that are involved in the places where they are deployed [2,3]. Structural engineers know that simulations that are based on the frequency domain provide fairly fast results compared to those that are based on the time domain; however, the high computational requirements of time-domain simulations provide highly accurate results [4]. From a practical point of view, time-domain simulations are not an option for long- and medium-term studies.

Note that structural failures in offshore aquaculture facilities generally involve high economical costs. It is not just about scheduling and executing dive operations to repair or replace buoys, nets, moorings, structural lines, connection nodes or anchors as each incident can totally or partially affect the cage structures and, therefore, the life that is contained within them [5,6]. For example, the breakage of a mooring line or the loss of a buoy can lead to the deformation of the supporting cage [7]; the volume of the cage is then reduced and the welfare of the population inside the cage is substantially diminished to the point of causing heart attacks among the individuals in the population due to high stress [8,9]. Another example is an open or broken net: wildlife can still enter the cage but the contained individuals can also escape to freedom. The objective of the incoming fauna is to consume the fish that live inside the cage, so damaged nets can result in big environmental problems due to the uncontrolled release of the individuals from inside the cage. In all cases, in addition to the costs of repairs, the large amounts of money that can lost due to the escape or death of the fish must be considered.

Although the objective of the existing literature studies has been to determine the forces that are involved in offshore aquaculture cage systems, the water flows in all of those studies were assumed to have a constant velocity in order to simplify the calculations and experiments. In this sense, the data that are provided by the current measurement devices are still presented as average speed. Moreover, this averaging procedure filters out the behavior of tides and waves by removing most of the frequency information. Knowing this, structural engineers add in a safety factor to allow for extreme cases. The greater the safety factor, the greater the anti-breakage guarantee and, obviously, the higher the construction and maintenance costs.

On the other hand, it is desirable to obtain values for the required parameters, such as tides and currents, that are as close to the real values as possible. In general, and as a first approximation, the existence of a harbor that is close to the location of an aquaculture infrastructure ensures the opportunity to measure these variables. In Spain, for example, aquaculture infrastructures are integrated into the sensor network of the Spanish Navy. The collected non-sensitive data are accessed using open source licenses from Internet repositories [10]. Equivalent data services also exist in other countries, such as the National Oceanic and Atmospheric Administration (NOAA) of the United States [11]. Obviously, the greater the distance between the port and the location under study, the lower the reliability of the data. In other words, the accuracy of the available data and, therefore, their validity is reduced the further away the facility is from a harbor. Then, for a location to develop new marine activity, the deployment of measurement equipment to obtain accurate data is required in most cases.

In the literature and industrial applications, there are five main approaches to measuring tides, currents and waves: radar, satellite imaging, turbine, tilt and Doppler techniques. The techniques that are based on satellite images and radar are only valid for measuring currents and waves on the ocean surface. In addition to their low resolution compared to the other methods, they do not allow for the measurement of deep-water currents [12]. Their main advantage is that they are remote sensing techniques, so they do not require local deployment. The other three techniques are used to measure the water velocity in specific locations.

A classic method for measuring tidal currents is the use of turbines. In this case, the instruments are oriented with the water flow and the rotation speed of the blades allows for the measurement of tidal current velocities. For example, the authors in [13] studied the importance of mooring systems and the interactions between the seabed, submerged buoys and the turbines. In that scenario, the focus of the paper was energy generation and the authors interest was in reducing the jerks in turbine thrusts. Since the behavior of the water is modified based on the measurement system, the use of turbines is not sensitive to transient events in the ocean. However, this method is still valid for measuring average speeds over long-term periods.

The most extended current flow meter is based on tilt measurements [14]. These instruments are based on sticks that contain inclinometers. The angle of the device determines the speed of the water current. Despite the usefulness of this approach, the main limitation is the construction principle of the devices. First at all, the measured cross-section of the water flow is the same as the stick length, on average. Secondly, the device must be attached to a fixed structure or seabed. It is well known that the ocean water flows reduce in speed near the seabed. There are other similar approaches in which the sticks are complemented with buoys instead [14].

On the other hand, based on the Eulerian method instead of the Lagrangian method, the existing literature has also presented Doppler solutions [15,16,17,18]. Using this approach, the speed of a water column can be measured. This solution is based on echolocation, i.e., pulses are emitted in specific directions and their reflections are sensed. In fact, this approach measures the speed of the particles that travel through the water column under study. It must also be taken into account that the detection of the particles depends on their size and the parameters of the emitted pulses (e.g., frequency, duration, amplitude, etc.) [16]. However, since there is no guarantee that the particles, wildlife or anything else that moves within the evaluated column is traveling at the same speed as the measured water, the obtained data may not be valid. Finally, the computational effort that is required to obtain the speed of the water is high compared to the other methods in the literature [18,19].

After reviewing the approaches and applications in the existing literature, we concluded they all require the measurement of tidal velocities. In this sense, the available tide measurement equipment provides the speed of the water under certain conditions, regardless of the used methodology. Whether the measurements are based on the Lagrangian or Eulerian methods, they all require the sampling of a medium or large set of data to provide the required values. As a result, medium- to long-term waiting times occur between the samples, from a couple of seconds to more than a dozen minutes. In other words, independent from the real sampling ratio, these instruments have low effective acquisition speeds. None of the instruments that were mentioned above provide information about the frequencies that are involved due to their own averaging concepts.

One possible solution that can override this limiting factor is to increase the sampling frequency of the available instruments. However, each piece of equipment has been designed at its maximum sampling frequency and the averaging methodologies focus on increasing their accuracy. Therefore, when it is possible to increase the sampling rate and reduce the averaging window, the performance of the instrument is compromised.

In summary, tide and current meters provide measurements of the speed of ocean waters and capture essential information that is required by designers and cage monitoring systems to guarantee the function and prevent the breakdown of offshore aquaculture infrastructures. This paper proposes the use of direct measurements of the acceleration and frequency of offshore ocean currents instead of averaged speeds. The key contributions of the paper are as follows:A novel instrument to measure the acceleration of tides and currents and their related frequencies;Experimental measurements of tides and currents that were obtained in the Canary Islands by deploying the fabricated prototypes at offshore aquaculture infrastructures, which were collated to the literature that has been published to date;From the obtained data, the domain component frequencies of offshore tides in short-, medium- and long-term measurements were acquired and the data for the three main axes were post-processed over 10 min (short term), one day (medium term) and a week (long term).

The rest of the paper is organized as follows. The next section presents an introduction to the principle of operation of the proposed instrument, focused on offshore aquaculture infrastructures and the behavior of tides and waves that has been previously described in the literature is also explained. Then, Section 3 presents the proposed instrument as an embedded system, from a design point of view. Here, the key points are detailed, such as scheduling, practical issues and the ultra-low-power design, in addition to performance.

In Section 4, the study location and the experiments that were performed to evaluate the proposed instrument are presented. This section starts with a discussion of the acceleration measurements, the influence of gravity acceleration and the sampling ratio issues, from a practical point of view. Then, based on the real acquired data from the offshore facility, the short-, medium- and long-term results are analyzed and discussed in terms of the time and frequency domains. A comparison to similar instruments that have been discussed in the existing literature is presented in Section 5. In the last section of the paper, our conclusions are provided.

## 2. Materials and Methods

### 2.1. Principle of Operation

Our target was to develop an ultra-low-power and long-term water current flow meter that was based on accelerometer. Based on the drag-tilt principle for a pendulum and assuming that *F_W_* was produced by a water flow, the angle *θ* was a function of the water speed where the device was submerged, as shown in Figure 1.

However, in order to measure this angle correctly, the rod of length L had to be rigid. Therefore, the measured velocity was a function of the speed of mass M and the stick or wire that was used as the rod. On the other hand, our target was to measure the water forces or acceleration. When we fixed a triaxial accelerometer to the mass, we could measure the tangential and centripetal acceleration of the pendulum movements, which we denoted as *F_T_* and *F_C_*, respectively. We did not forget the gravitational force *F_g_* and, because the scenario was underwater, the buoyancy force *F_B_* also had to be considered. Their module values were:(1)$$F_{g}=m\times g,$$
(2)$$F_{B}=\rho \times g\times V,$$
where *m* is the mass of the submerged device (kg), *g* is the gravitational acceleration (m/s^2^), *ρ* is the density of the salt water (kg/m^3^) and *V* is the device volume (m^3^).

We also assumed that the pendulum was in equilibrium (see Position 1 in Figure 1). The force system was formulated as:(3)$$T_{lb}=F_{g}-F_{B}=g\times \left(m-\rho \times V\right)$$

That is, the tension force *T_lb_* that was supported by the rod was the gravity force minus the buoyancy force. This equilibrium state defined the lower bound for the range of tension forces that were needed to support the rod. Equation (Equation 3) defines this behavior in terms of the floatability of a submerged mass: a positive value indicates that the mass would float towards the ocean surface, while a negative value indicates that the object would fall to the seabed. Due to the mass being attached to a rod in our scenario, the behavior of positive values was buoyant and that of negative values was pendulous when the tension force was positive.

Outside of equilibrium (see Position 2 in Figure 1), the equation was as follows:(4)$$F_{T}+F_{C}=F_{W}+F_{g},$$
where *F_T_* and *F_C_* are the tangential and centripetal forces, respectively, *F_W_* is the water force and *F_g_* is the gravitational force. Of course, when in equilibrium, *F_C_* was the gravitational force *F_g_* and *F_W_* and *F_T_* were zero.

### 2.2. Deployment Considerations

Equation (Equation 4) was used to model the forces that were involved in the behavior of the proposed tidal current meter. Because this device was intended for use near offshore infrastructures, the features of the ocean currents had to be taken into account. These considerations were fed back from their physical values, which allowed us to understand the composition of the forces and the complexity of the equation.

The ocean surface is seen as a succession of irregularly distributed peaks and valleys. In the literature, researchers have described ocean waves in terms of their regularity [20]. For example, purely irregular ocean waves are produced by unusual events, such as submarines, earthquakes, tsunamis or objects falling into the ocean (e.g., meteorites). In addition, shallow waters with irregular seabeds or extreme climatic events (e.g., hurricanes) also produce irregular waves.

However, regardless of the ocean wave complexity, waves can be described using the Fourier model, in which the ocean surface elevation *η*(*t*) is the composition of several sinusoidal functions [21], i.e.:(5)$$\eta \left(t\right)=a_{0}+\sum_{i=1}^{N-1}a_{i}sin\left(\omega_{i}t+\phi_{i}\right),$$
where *a*_0_ is the averaged level of the ocean surface and *a_i_*, *w_i_* and *ϕ_i_* are the amplitude, frequency and phase of each sinusoidal *N* component of the model, respectively. The simplest cases of regular waves are modeled with single sinusoidal functions. When ω = 2*πf*, the ocean wave period is *T* = 2*π*/*ω*. The expected function for *F_W_* in Equation (Equation 4) was in the same form as its function in Equation (Equation 5).

On the other hand, there is a basic component that is always present. Any large concentration of water, such as an ocean, is affected by the gravitational forces of the moon and sun. Gravity moves ocean water to restore gravitational balance. In the case of deep water, this movement follows a harmonic behavior. Every 6 h, 12 min and 30 s, there is a maximum or minimum value in terms of ocean surface level, which are called high tide and low tide, respectively. In terms of Equation (Equation 5), the gravitational wave period *T* was equal to 12.417 h.

Locally, the ocean surface is forged by the wind creating waves. Depending on the applied forces, a wide variety of different types of waves can be created. Regardless of the wave shapes that are formed, it is well known that water currents that are caused by wind are attenuated with depth. All of the related research literature has considered the influence of waves that are generated by the wind to be negligible from a depth (*d_n_*) that is greater than half of the wavelength *l_w_* (see Figure 2 for more detail).
(6)$$d_{n}=l_{w}/2$$

This value is used as a frontier for considering shallow or deep water. In summary, the behavior of shallow water depends largely on the seabed and the climatic conditions, such as wind, pressure, temperature, etc., in addition to gravitational waves. In the case of deep water, the behavior depends mainly on gravitational forces, temperature and salinity. In general, the literature has considered deep water to be from 7 to 10 m.

In addition, it is also well known that there is a limit to the height of a wave before it breaks [22,23]. When a wave reaches a height that is greater than 1/7 of its length, the wave collapses.
(7)$$h_{w}\left(break\right)\geq l_{w}/7$$

Finally, ocean waters are not quiet. Nowadays, the ocean circulation theory is generally accepted [24]. Mainly due to the forces of the moon and the sun (but without forgetting the contributions of the wind that is applied to the ocean surfaces), the salinity and temperature of deep water and the continental land masses, among other factors, produce the regular circulation movements of the oceans. Figure 3 depicts the most important ocean currents. The consequences of this circulation are that the oceans are in constant movement across the globe and that they always maintain the same direction [25].

Nowadays, it has been confirmed that the seasons and extreme weather events, such as storms, can modulate global ocean movements but never change the direction of tidal currents. It is noteworthy that there are some studies on the influence of climate change phenomena within the literature that have assessed the variations in ocean currents that could occur in the not too distant future.

### 2.3. Offshore Aquaculture Infrastructures and Their Location

By definition, offshore means far away from land. In this sense and considering the facts that were discussed in Section 2.2, disturbances in ocean behavior that are due to jagged coastlines, irregular seabeds or shallow water are not present in the forces that act on offshore infrastructures.

Humanity needs more and more food resources due to population growth. The food industry has come to see aquaculture as a new frontier to explore the expansion and provision of food resources. From an ecological point of view, the aquaculture industry has also become seen as a way to reduce the use of traditional fishing techniques that deplete life in our oceans. Both of these reasons have contributed to the exponential growth of the aquaculture industry over recent years.

The visual impact of offshore aquaculture infrastructures is notorious. In this sense, the development of these infrastructures not only has to comply with government regulations regarding the environment, but it also has to enter into direct competition with other deeply rooted industries, such as tourism, artisanal fishing and port infrastructures. Although their presence on the surface of the ocean draws attention, only a small part of the entire infrastructure can be seen [26] as the occupied volume stretches from the surface to between 25 and 50 m deep (see Figure 4 for more detail). As a consequence, these semi-submerged installations are affected by the wind in the first instance and then secondly by the waves and currents of the so-called shallow water. Finally, since there are parts of aquaculture infrastructures that are in deep water, deep-water currents also have to be considered (see the right side of Figure 4b) [27].

The ideal place to deploy this kind of infrastructure is a marine area that is protected from strong tides and wind, such as a bay or beach; however, these sites are already occupied by other industries, such as tourism or port infrastructures. A quick and easy solution would be to place new aquaculture facilities between 1 km and 3 km away from the coastline. In general, this range of distance provides locations at which shallow-water currents do not depend on the seabed or the shape of the coastline. In other words, most of the turbulence that occurs in foreshore areas is not present at this distance. Furthermore, in most cases, this range of distance is considered to be within the open sea zone, with the infrastructures being placed on continental shelves. Distances of greater than 3 km must be studied in detail since water depth increases dramatically after the edge of the shelves down to the abyssal plains.

Nowadays, industrial technologies exist that can operate without any problems at depths of up to 2.5 km, e.g., the oil extraction industry. However, the greater the depth, the greater the costs, which have to be justified by high profits. In the case of the offshore aquaculture industry, deployment and maintenance is performed using traditional marine tools to control the expense and be cost-effective, i.e., the use of anchors, buoys, ropes and nets. So, the practical maximum depth of deployment is limited to 35 to 150 m, which corresponds to 1 to 3 km from the coast.

## 3. Prototype Design

### 3.1. The Measurement Unit

This proposal was based on the use of a microcontroller and an accelerometer. Figure 5 shows a basic block diagram of its architecture. The entire system was managed by an ultra-low-power single-core microcontroller (the MKL17Z256 model from NXP). This integrated circuit included a flash memory for programs and data of 256 KB and 32 KB of RAM, respectively. It was directly connected to an accelerometer and a micro secure digital memory (uSD). We used MMA 8451Q the digital accelerometer, which it had three 14/8-bit axes.

It was mandatory to include an energy accumulator to allow the measurement unit (MU) to work in stand-alone mode. The final purpose of the MU was to be integrated into an underwater sensor network [26]. However, in this paper, we just used its communication unit to program the experiments, download the measured data and charge the energy accumulator.

Since the communication and charging processes had to be tested underwater and in the laboratory, the communication technology and frequency had to be selected according to the water conditions. In our case, we included a low-frequency identification (LF-RFID) communication unit.

The total cost of the instrument was under USD 50. The microcontroller, the uSD and the battery costs were around USD 8 each and the capsule costed USD 12. The printed circuit board with gold-plated layers cost around USD 4 and the spare parts and soldering cost around USD 50.

### 3.2. Practical Issues

For a MU that is deployed in the ocean to measure acceleration where there are no acting forces except gravity, the MU X direction has to be parallel to the gravity vector. In the same scenario, the plane that includes the Y and Z directions is parallel to the ocean surface. Therefore, the X direction is also normal to the ocean surface. Finally, the acceleration of X is negative downward (toward the seabed) and positive upward (toward the ocean surface). For this reason, in this study, we identified the X direction as the vertical direction.

Following this nomenclature and from a practical point of view, other considerations also had to be taken into account. The first was the so-called misalignment. The arrangement of the axes is closely related to the positioning of the integrated circuit (IC) acceleration measurement device. Although the orthogonality of the measured axes is always guaranteed at the IC level within its package, it is mandatory to check the misalignment of the measurement circuit with respect to the complete MU capsule.

In general, the sources of error are due to the misalignment of placement and the soldering steps of printed circuit board (PCB) construction, e.g., handmade processes, automatic processing without calibration and excessive uncertainty ranges. The internal support structures that hold the battery and printed circuit board are also sources of misalignment errors. In this second case, the fixing of each element inside the MU capsule must be guaranteed; otherwise, a new error could be produced by the semi-free movement of parts inside the capsule and would have to be taken into account.

By ensuring each part is fixed inside the MU capsule, the misalignment error can be canceled out by eliminating the error as a regular offset error. Therefore, in each MU assembly process, the internal elements must be fixed before sealing the capsule so that the misalignment offset can be obtained.

On the other hand and from a practical point of view, the sampling frequency that is used in the MU is another important issue. Since we were dealing with signals that had their own bandwidths, we used the Nyquist theorem to define the lower bound of the MU sampling rate. It is well known that the higher the selected frequency, the better its reconstruction and post-processing. However, our MU was an ultra-low-power system, so increasing the frequency led to higher power consumption; therefore, we established a trade-off between the energy consumption and the sampling frequency.

In our experiments, we chose a sampling frequency of 12.5 sps (samples per second). Based on the literature, we expected the maximum bandwidth of the acquired data to be 3 Hz [28,29]. This value represented more than twice the maximum expected frequency.

### 3.3. Scheduling

The most common deployment for this type of instrument is in stand-alone/autonomous mode, in which the instrument is isolated from external power sources and it executes the measurements using energy that is provided by a battery.

Since the energy is limited, scheduling is of great importance and the measurements have to be programmed for when the variable under study is remarkable. In this sense, the scheduling defines how the acquisitions (bursts) have to be and how much time the instrument has to wait until the next burst.

The measurement procedure for an offshore aquaculture infrastructure consists of three steps. The first is to configure and install the instrument (deployment). In this step, the user programs the required settings into the instrument, such as the sampling ratio, the maximum speed, the burst period, the waiting time between bursts, etc. Then, in the second step (experiment), the device measures the water currents following the programmed instructions. The last step consists of retrieving the instrument and downloading the acquired data (retrieval and download).

Figure 6 presents these three steps, in addition to a block diagram of the execution of the instrument under study. On the right of this figure are the working modes of the microcontroller (CPU) and the accelerometer (ACC). The actions that are presented in this block diagram were as follows:Program: In the beginning, the microcontroller and accelerometer were active in run mode by default and the acquisition parameters needed to be specified, so the instrument was connected to a personal computer (PC) (which also charged the battery) and once the user had finished programing, the system went into Hibernate_1_;Hibernate_1_: The CPU was in ultra-low-power (VLPS) working mode and the ACC was completely turned off, so a time had to be provided for the deployment step (t*_D_*), for which diving operations had to be carried out;Setup: The CPU woke up using its low-power wake-up timer (LPWUT), configured the ACC and started the acquisition of acceleration measurements;Measure: The ACC acquired the acceleration measurements using its internal 32-sample first in first out (FIFO) buffer (note that the ACC did not require CPU support, so the CPU went into in VLPS again);Cache/Store: Once the ACC FIFO was close to being full, a hardware interruption was generated to wake up the CPU. Initially, all FIFO data were temporally stored in the microcontroller RAM (we used 24 KB of RAM memory as the intermediate memory). When it is necessary acquire more acceleration measurements and the intermediate memory was not full, the inner loop executed a new measurement cycle. When this intermediate memory was full, its content was stored on the uSD and the inner loop implemented the burst acquisition mode;Hibernate_2_: The CPU went into VLPS and the ACC turned off, which implemented the delay between burst acquisitions using the outer loop and then the CPU woke up using its LPWUT;Hibernate_3_: Once the system finished the acquisitions, the CPU went into VLPS and the ACC turned off. After a set amount of time (t*_R_*), the device was retrieved from the deployment location and the acquired data were downloaded to the PC.

**Figure 6 sensors-22-05513-f006:**
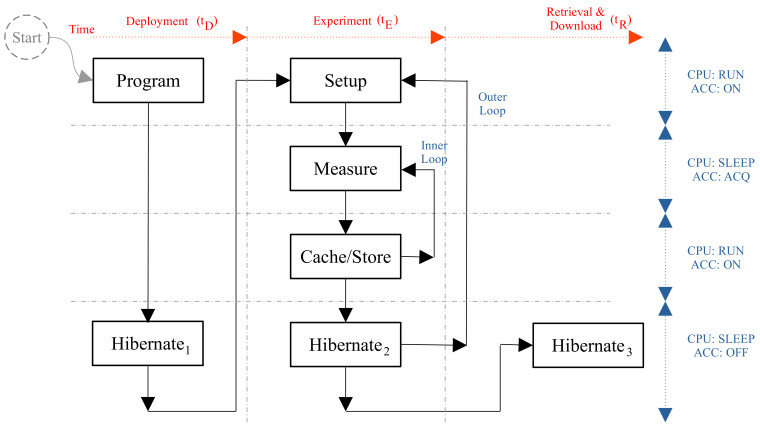
The execution schedule that was implemented for the proposed deep-water current meter.

### 3.4. The Ultra-Low-Power Design and Instrument Performance

Despite the system being intended to operate in ultra-low power, there were some key design points regarding the CPU and ACC usage that need to be explained. Instead of conservatively optimizing everything in the design, we preferred to use an overview of the system to focus on the optimization of resources that had high consumption costs. From the point of view of power consumption, the proposed instrument had three obvious main power consumers: the CPU, ACC and uSD.

The selected CPU had several working modes, from a high-performance mode to an ultra-low-power mode. The more of its internal modules that were turned off, the lower the power consumption. In this sense, we needed to use at least a run mode and a sleep mode. The selected run mode have a maximum working frequency at which it produced ultra-low-power behavior. Increasing the frequency over this limit increased the power consumption substantially. We did not select a lower frequency because we wanted to ensure a fast data transfer between the three main components. The CPU working frequency was 8 MHz in its very low-power run mode (VLPR). The CPU power consumption in this mode was under 500 μA.

On the other hand, the selected sleep mode was the so-called very low-power stop (VLPS) mode. In this mode, all of the microcontroller modules were disabled, except the internal clock, the external interruption module and the RAM. The MKL17Z256 microcontroller consumed under 1 μA of power.

Due to turning off the ACC, its consumption was zero when not used. When the ACC was powered, it went into stand-by mode and consumed 1.8 μA of power. Obviously, the consumption of the data acquisitions depended on the specified sampling frequency. In our case, this consumption was about 6 μA. The instruction manual for the ACC did not indicate what the power consumption was when it was in data transfer mode. The reason for this was that the consumption depended on the values of the data. The bus transfer was a I^2^C compliance and a zero in this bus produced the short to ground of a 4.7 kΩ pull-up resistor. The clock line worked in the same way. When we assumed a half of zeroes and ones in the data line transmission, the consumption was 0.574 mA for a power supply of 2.7 V.

Finally, the uSD required at least 2.0 μA when it was powered and this non-volatile memory required 27 mA to store the data.

Based on the currents that are shown in Table 1, it was easy to determine that the uSD was the maximum power consumer. The minimal information in a uSD memory is called a block and the block length is 512 bytes. Due to each sample being measured by a 14-bit accelerometer with three axes, every sample required 3 × 16 bits of memory (6 bytes). The 24-KB RAM of the CPU was used as intermediate memory to accumulate the sampled data from several consecutive 32-sample ACC FIFOs and send a block to the uSD as fast as possible.

With a sampling frequency of 12.5 Hz, the ACC FIFO was full in 32/12.5 = 2.56 s. On the other hand, the CPU intermediate memory was full in 327.68 s. With a SPI data transmission of 200 KHz, the intermediate memory was stored in the uSD in approximately 1.9 s. The writing time for a Class-2 uSD is 2 MB/s, so the uSD writing process only required 12.23 ms, but the writing process was extended to 1.9 s due to its data bus transfer requiring 1.9 s.

Then, we were ready to estimate the total consumption of our system, as presented in Figure 5. We assumed the worst-case scenario, which consisted of acquiring acceleration measurements continuously without any hibernation time.

Table 2 presents the currents and times that were required in this worst-case scenario. The total energy required was 1.56 μAh. In our prototype, we used a Panasonic NCR18650B battery (lithium-ion, 3.7 V and 3400 mAh) as the power source. For this case, the runtime of the experiment was 188 days.

The deep-water tidal meter that was developed in this research was based on the direct measurement of the acceleration vectors of water currents and their frequency components instead of the use of averaged water speeds. Since we used a commercial accelerometer, the performance of our instrument was the same as that of the selected accelerometer. Based on the instruction manual of the accelerometer for our configuration, the resolution was g/4096 = 0.2441 mg = 2.39 mm/s^2^, the working range was ±2 g = ±19.6 m/s^2^ and the measured output noise was 3.4 mg/$\sqrt{Hz}$.

## 4. Experiments

All of the experiments in this study followed the same procedure. The first step was to program our proposed MU to obtain the raw acceleration data for a specified time period. Since our lab was located on land and far from the deployment location, the start of the experiment had to be delayed. Another consideration to keep in mind was that the deployment had to fit in with the other diving operations that were scheduled at the offshore aquaculture infrastructure. In general, each experiment was delayed by at least 48 h to provide time margins for the divers.

Once the experiment began, the raw acceleration data were continuously captured at a fixed sample rate and stored on the on-board micro secure card (uSD). As soon as the experiment was over, the MU went into ultra-low-power sleep mode to wait to be recovered. Finally, with the MU back in our lab, we extracted the uSD and copied the acquired raw data onto a PC (Windows 10, i7-4930 K 3.4 GHz and 64 GB of RAM). All data that are presented in this paper were measured using LabView 2021 from National Instruments.

### 4.1. Study Location

To demonstrate the utility of our approach, we deployed several prototypes of our MU in a real scenario. The aquaculture facility that was used is the property of Aquanaria S.L. in the Canary Islands (Spain). Figure 7 shows the location of the Canary Islands in the Atlantic Ocean. Figure 7b,c show two Sentinel 2 satellite photos (in the RGB bands): one of the island of Gran Canaria (GC) and one of an enlargement of the area in which the offshore aquaculture facility is located.

The exact placement of the offshore facility is 27°46′28″ N and 15°28′23″ W and it is located about 2 km away from the coast. This facility is located in the middle of a 1-km wide semi-flat seabed with a bathymetry that starts at 25 m and ends at 50 m. The seabed around the offshore facility is 35 m deep. The seabed in the occupied area does not have an irregular form. The offshore facility is composed of two independent arrays of 2 × 6 cages. The devices were deployed on the two mooring lines that were the most exposed to the water currents at 15 m of depth.

### 4.2. Raw Data

In this study, a first experimental approximation to measure the tidal currents in terms of acceleration was proposed to obtain data for a short period of time and to analyze those obtained data. Figure 8 shows 20.48 s of the raw data that were captured on 5 May 2021 at 15:44 at the study location, which was described in Section 4.1, at a depth of 15 m. This time period corresponded to 256 acceleration measurements. The image shows the magnitude of the acceleration vector ACC_XYZ_ and its three orthogonal axes (ACC_X_, ACC_Y_ and ACC_Z_) in terms of gravitational acceleration g (9.81 m/s^2^).

During this period, the magnitude (see ACC_XYZ_ in Figure 8) was close to the unit value. The range of accelerations was [0.876, 1.078] g. When we took into consideration that the gravitational acceleration was 1 g, the variation range of the measured magnitude was basically [−0.124, 0.078]. Considering that the magnitude was computed using the Pythagorean theorem, its variation was lower than the variations in its components in most cases.

On the other hand, when we looked at ACC_X_ (vertical direction), we could see that the measured values were close to unity. This implied that the MU was in a vertical position most of the time. In other words, the tidal force had low values that were close to zero. As expected, the time domain signal that is presented in Figure 8 exhibited a continuous curve, with several maximums (peaks) and minimums (valleys). At this point, it was possible to detect the periods/frequencies of the deep-water wave signals that were involved using a peak-valley detector [30].

Since we were looking for the periods/frequencies of ocean waves in deep water, it was interesting to study the signals in terms of frequency. Figure 9 presents the previously reported raw acceleration data in terms of frequency distribution. In this experiment, we first captured the raw acceleration data using our MU at the location that was described in Section 4.1. The acquired data were stored in the on-board micro secure card (uSD) and downloaded onto our lab PC to be processed using a fast Fourier transform (FFT) algorithm in LabView 2021 (National Instruments).

Because the FFT algorithm depended on the length of the processing data, we used all of the possible lengths between 8192 and 128 in powers of two, as shown in Figure 9a–g. Due to our MU using a sampling frequency of 12.5 sps, the raw data corresponded to times from 655.36 s (10′55.36″) to 10.24 min.

The arrangement of Figure 9 follows a matrix organization. The first column of graphs shows the vertical direction and the other two columns present the other two acceleration components (ACC_Y_ and ACC_Z_, respectively). Each row of graphs represents a length of acquired data. In this way, Subfigure (a.1) depicts the FFT of the vertical direction with a length of 8192 samples and Subfigure (b.1) depicts the same acquisition direction but with only 4096 samples. Finally, Subfigure (g.3) presents the ACC_Z_ FFT with only 128 samples.

The amplitude of the signals decreased to zero as the frequency increased. Although the maximum frequency that was obtained based on the Nyquist theorem was 6.25 Hz, we only represented up to 3 Hz because the obtained signals that were above 2.5 Hz were the baseline noise of the system, which was close to zero.

At first glance, regardless of the axis and frequency that were studied, the presence of noise was very noticeable. The reason was quite simple: since tidal currents are randomly generated, the acquired signals were also random. However, the literature has described that a wave or tidal current with a period/frequency of t*_w_* = 1/f_*w*_ would have an amplitude that is modulated. This modulation in deep water is half of the original frequency. Therefore, its frequency decomposition has a fundamental harmonic and the resulting component of a created wave. This effect is called group velocity.

Figure 9a.1 and a.2 clearly present the AM modulations at 0.25 Hz, 0.6 Hz and 1.3 Hz. The power differences between these modulations were also significant, with 30 dB and 20 dB between them. Furthermore, the FFT of ACC_Z_ that is presented in Figure 9a.3 was above the modulation at 0.25 Hz to 0.15 Hz. This behavior was identical to the behavior that is shown in the other subfigures.

As the length of the FFT decreased, the resolution became worse. Obviously, the required computational effort was less with the smaller lengths. In addition, this reduction produced less spectrum power and the tendency of the modulation lobes was to disappear. In the case of fundamental/carrier frequencies, this behavior was not as pronounced, so as the transform length reduced, the carrier stood out from its lobes. In terms of frequency, we could conclude that it was possible to obtain very precise measurements of the frequencies of fundamental waves at a low computational cost using reduced FFT lengths. However, the measurement of the modulation lobes required large FFT lengths.

### 4.3. One Day

By evaluating the raw data that were obtained for short periods (≤10.24 min), the following experiment tested the behavior of the acceleration over a full day. Figure 10 shows the acceleration measurements of the three orthogonal axes for the one day. The acquisition began on 5 May 2021 at 12:00 noon. In addition to the acceleration measurements, we also present the sea level, as measured at the Las Palmas 2 port (coordinates 28°8′32.78″ N and 15°24′37.00″ W), which is also named Faro [31]. The tide gauge was far away from the infrastructure at 3′46″ S and 22′46.48″ W, which represented a linear distance of 41 km.

Figure 10a–c represent the range of measured acceleration values and the average acceleration values for ACC_X_, ACC_Y_ and ACC_Z_, respectively (they are referred to by their left ordinate). In addition, the sea level that was measured by the Faro tide gauge is also presented in meters (right ordinate).

As indicated in Section 2.2, the maximum and minimum sea levels occur approximately every 6 h. Since we were acquiring measurements over a full day, there were two maximums and two minimums. In particular, Figure 10 shows the vertical direction of the MU, i.e., ACC_X_. Please note that for this direction, a measurement of close to one meant that the MU was vertical. We could clearly observe the two maximums and minimums in the acceleration measurements. In addition, there was a delay between the peaks and valleys.

Remember that the maximum value is when the water accumulation is within the zone of the maximum attraction of the earth to the moon and the sun. To reach the maximum value, the tidal currents must move the water until that value is obtained. Once it is reached, the tides stop pushing by the same magnitude. Since we were in deep water, the water current never reversed direction due to the circulatory movement of the ocean.

Figure 10a shows the behavior that was described above. The vertical movement of the MU reached its maximum 4 h after starting the full-day experiment, which started at 12:00 noon on 5 May 2020. Then, the MU unit became vertical around 15:00 on the same day and then stayed in more or less the same position for 2 h. After this time, the gravitational tidal forces acted to reach high tide. The beginning of the vertical movement of the MU was perfectly synchronized, in temporal terms, with the measured low tide on that day at that time. About 12 h later, a similar event occurred, as is shown by the ACC_X_ measurements.

The same behavior was present in the measurements from the other two axes. While ACC_Z_ followed the same tendency as ACC_X_, ACC_Y_ was similar but reversed. The ranges of movement that were recorded when the axes were parallel to the ocean surface were greater than those when the axes were vertical.

While the movement of shallow water flows in one direction and follows circular patterns due to surface waves, the movement of deep water follows the classical theory of fluid motion, i.e., the water flows in one direction and when there are fluctuations, they are due to irregularities on the seabed, other new incoming forces or temperature changes. As previously discussed in Section 2.1, we expected the measured values to include gravitational acceleration and tidal forces. Since we did not expect vertical currents in the offshore area, we expected to measure forces that were parallel to the ocean surface.

When an incident force *F_w_* that was parallel to the ocean surface was applied to our MU, based on Equation (Equation 4), the resultant tangential and centripetal forces were:(8)$$F_{C}=F_{w}\times cos\left(\alpha \right)+F_{g}\times sin\left(\alpha \right),$$
and
(9)$$F_{T}=F_{w}\times sin\left(\alpha \right)+F_{g}\times cos\left(\alpha \right).$$
where *α* is the angular difference between the normal vector to the ocean surface and the incident force on the plane containing both vectors.

Considering that the MU position was close to the vertical position most of the time, based on Equations (Equation 8) and (Equation 9), the measurements were more sensitive/variable on the Y and Z axes than on the X axis (i.e., *F_T_*). For this reason, the gravitational acceleration is more visible in Figure 10b,c.

On the other hand, Figure 11 illustrates the same raw data that were captured throughout a full day from the point of view of frequency. In this case, the length of the FFT was 2^20^ = 1,048,576 samples, which represented 23 h, 18 min and 6.08 s. Therefore, just under two complete tidal cycles were captured.

At first glance, the 1.5 Hz bands disappeared at the noise baseline compared to the 10-min FFTs (see Figure 11 for more detail). However, this conclusion was wrong. In this scenario, for a given sample rate, increasing the length of the FFT improved the visibility of repetitive signals and reduced single-event signals. Therefore, a large FFT was basically a probability distribution function. So, the 1.5 Hz signal was there but its probability of occurrence was very low compared to the other frequencies.

Then again, the gravitational tides had a period of T = 44,701.2 s, which corresponded to a frequency of 22.37 μHz. Each subfigure also includes an enlargement of the ultra-low-frequency range of the FFTs, in logarithmic scale, from 1 × 10^−5^ to 0.01 Hz. In this sense, for all of the MU axes, the enlargements in the subfigures show a frequency peak at 35.76 μHz, which represented a tide with a period of 7 h 46 min and 4.2 s. Obviously, this frequency was not the gravitational tide.

This was the other issue with the selected FFT lengths and sampling frequencies. Following the Nyquist theorem, a sample rate of 12.5 sps allows for a 6.25-Hz sinusoidal signal to be correctly reconstructed. In this sense, an FFT that was calculated with an input signal of 2^20^ samples had a resolution of 12.5/2^20^ = 11.92 μHz. Assuming an error of ±1 bin, the figure value was OK; therefore, there was a resolution problem. In order to obtain a 0.1-μHz resolution for a given sampling frequency, the length of the FFT was computed as:(10)$$length\left(FFT\right)\geq 2^{Log\left(\frac{SamplingFrequency}{resolution}\right)/Log\left(2\right)}=2^{Log\left(\frac{12.5\;Hz}{0.1\mu \;Hz}\right)/log\left(2\right)}=2^{26.89}$$

Assuming the value of Equation (Equation 10) was the length of the FFT, the resulting time for a sampling frequency of 12.5 Hz was 124.27 days. On the other hand, the samples needed a memory of 6 bytes/sample × 2^27^ samples = 768 MB. Regardless of the computational effort to run an FFT of this length, this memory was not practical from an ultra-low-power microcontroller, such as the device that was embedded in our MU.

According to Equation (Equation 10), an easy way to reduce the number of samples that were required to obtain the desired resolution was basically to reduce the sampling frequency. However, this solution was not possible because the 12.5 sps rate was necessary to measure the maximum possible wave frequency, based on [29].

### 4.4. One Week

Figure 12a–c present the continuous measurements for a full week (168 h) using a sample rate of 12.5 Hz for the three orthogonal acceleration axes of our MU. These figures also include the sea level for the capture dates on the right-hand axis. The experiment started on 5 May 2021 at 12:00 noon and ended on 12 May at the same time.

In all of the figures, the gravitational tide is clearly present. Just before high tide, the acceleration values were at their maximum. After high tide, the acceleration values progressively reduced. The reverse behavior occurred at low tide. The total number of samples from the full week was 7.46 Msps. In terms of sensitivity, as with the 10-min or full-day records, the variability of the movements was greater on the Y and Z axes of our MU, which were measured parallel to the ocean surface.

The periodicity of the gravitational tides means that the high and low tides are not identical every day, i.e., despite the fact that there are high and low tides approximately every 6 h every day, the levels that are reached are not the same. This was the reason that different waveforms were observed over time in the acquired acceleration data.

For example, Figure 12a shows the measurements of ACC_X_ and presents the acquired data from the first 40 h, during which the behavior was similar. Over a 1-month period, the high and low tides start close to the mean ocean level and then increase their distance from the mean ocean level. In terms of sea level, this period corresponds to the sea level rise throughout the month. Over the next 40 h, the high tide reached its maximum level of gravitational attraction. The peaks increased to their maximum and the valleys stayed close to the mean ocean level. Then, over the next 40 h, the peaks stayed approximately the same and the valleys decreased to their minimum. Finally, for the rest of the week, the maximum and minimum values stayed the same.

As discussed in Section 4.3, the extraction of the gravitational tide frequencies from the acquired raw data was not practical using the FFTs. In this study, we used a peak-valley detector to reduce the sampling frequency to 5 min instead of 12.5 sps. The reduction in this value was applied by averaging the values. To meet the requirements for the practical implementation of our proposed MU, as soon a sample was acquired at 12.5 sps, this value was accumulated with the others from the next 5 min.

## 5. Comparisons

Table 3 presents a comparison between several current commercial and research approaches. In general, the Doppler-based techniques need to be installed on the seabed, whereas ours could be installed on a mooring line. The other solutions are based on floating buoys. It is necessary to highlight that these instruments use the so-called internal sampling frequency to measure water currents, which does not coincide with the sampling frequency that was observed in our output data (see the internal *F_s_* and *F_s_* values in Table 3). The exception is turbine-based instruments, for which a pulse is produced for every 360° rotation of their blades and there is no sampling frequency as such. This difference in frequency is necessary to preprocess the sampled data and increase the original performance using averaging or other filtering techniques.

The *F_s_* was in the range of a couple of minutes to 64 Hz in the studied devices. The most common value was around 1 Hz and was 12.5 Hz for our instrument.

Note that some Doppler devices are general purpose instruments and the sampling frequency is increased to 64 Hz. This is because other application specifications are other objectives of this type of equipment that need to reach greater object speeds, such as shallow water measurements, the deformation of submerged structures, biomass movements and river flows. In those other applications, the expected speeds are greater than those in deep-water applications. In our method, the maximum speed measurement was based on the average value of the tilt-based devices. On the other hand, our accuracy was better than that of the studied devices. The resolution that was obtained using our method was equal to the best Doppler solution (0.01 cm/s).

Most of the solutions use a micro secure digital memory card as the memory to record the sampled data. However, some industrial solutions exist that include internal memory that is not upgraded. Some devices include an Internet server connection, but this communication advantage introduces several disadvantages in terms of runtime and size.

The size, geometric form and weight of the instrument play an important role in the measurement methodology, in addition to the size of the battery. Our proposed instrument had the smallest volume. In Table 1, the costs of the devices range from USD 20 k to USD 50–40. Our instrument was in the lower range.

In general, the power consumption and energy sources determine the stand-alone runtimes of the instruments. The stand-alone runtimes in the table were obtained when the instruments were using their own batteries as the power sources. In our case, the instrument could obtain continuous measurements for up to 180 days at its maximum performance.

Since most of the applications run in stand-alone mode, it was very important to evaluate their power consumption efficiency. The research literature and commercial solutions have not presented this data. However, we could compare the total number of measurements that were obtained under the same conditions as an energy efficiency figure of merit (FoM). The column labeled "Number of Samples" in Table 3 presents the FoM values, which were calculated as the runtimes in stand-alone mode multiplied by the sampling frequency (*F_s_*).

Taking the most accurate and high-resolution Doppler instrument [34] as our reference, the other Doppler solutions were more efficient in terms of the number of captured data by up to three orders of magnitude (see reference [33] on the second row). The reason for that was the resolution. The most efficient Doppler solution had a lower resolution by up to two orders of magnitude. Similarly, the other tilt-based approaches were 1.55 to 2.8 better, but with a worse resolution. When we compared our approach to the best Doppler solution, our instrument was four orders of magnitude better with the same resolution and accuracy.

## 6. Conclusions

In this paper, an ultra-low-power embedded system was proposed to measure tidal currents at offshore aquaculture infrastructures using an accelerometer. The principle of operation was described in detail. The proposed method was implemented and used for recording raw data from the offshore Aquanaria S.L. infrastructure in the Canary Islands, Spain. The acquired data were studied to determine the dominant component frequencies of offshore tides in the short, medium and long term. These frequencies were tested over 10 min, a full day and a full week by acquiring the raw acceleration data for the offshore tides and storing the recorded data on a micro security disk card.

In summary, the raw acceleration data were studied in the time and frequency domains. The frequency study was based on the use of FFTs. The acquired raw data were processed off-board. The limited computational capabilities of the fabricated prototype introduced resource allocation as a design variable to determine the best algorithm to extract the fundamental frequencies of tides in offshore areas.

The selected sample rate of 12.5 sps allowed for the measurement of the fastest local currents, as well as gravitational tides. The fastest were below 3 Hz and the slowest were around 23 μHz. The range of frequencies to be measured required different techniques to be used in order to be efficient in terms of the memory and CPU effort in the ultra-low-power design.

This study found that the use of FFTs could be useful as a first approximation step to determine the range of the main frequencies that were close to the selected Hz value. For this range, it was better to use a peak-valley detector instead of FFTs due to the variability of the waves. In all of our experiments, the prototype demonstrated the usefulness of our proposed method for obtaining the profiles of the acceleration data of offshore tides.

From the comparison to current research and commercial approaches, our proposed methodology was able to attain the same error rate as the Doppler solutions (lower than 1%), with a resolution of 0.01 cm/s for a measurement range of ±0.8 cm/s. This ultra-low-power design allowed for continuously sampling (in burst mode) for more than 180 days using a common 18,650 lithium-ion battery compared to the other solutions that only allowed for no more than 56 days of measurements under the same conditions.

## Figures and Tables

**Figure 1 sensors-22-05513-f001:**
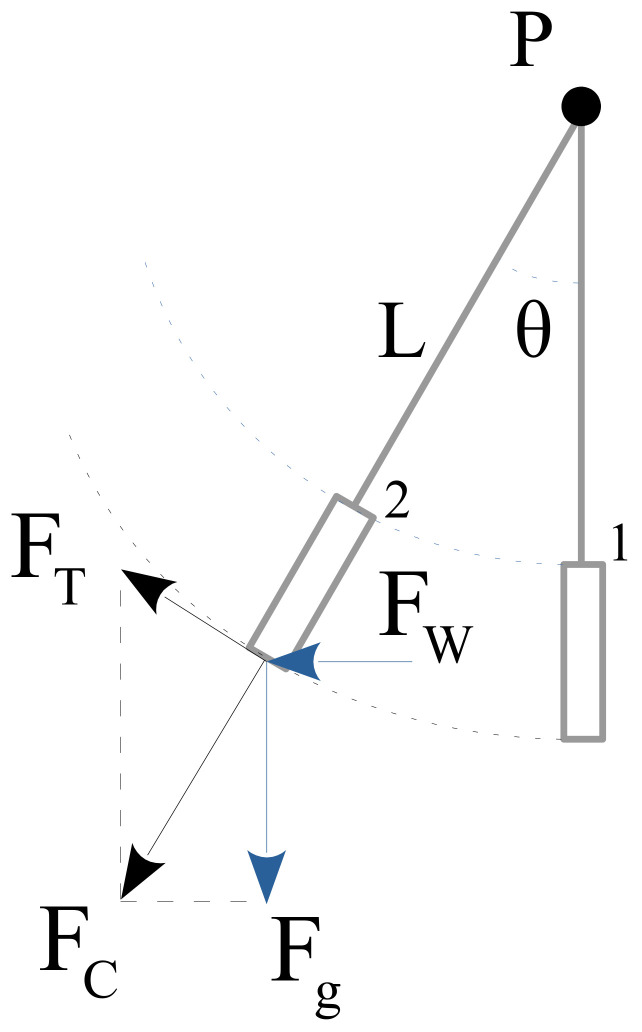
The measurement device and its basic theory.

**Figure 2 sensors-22-05513-f002:**
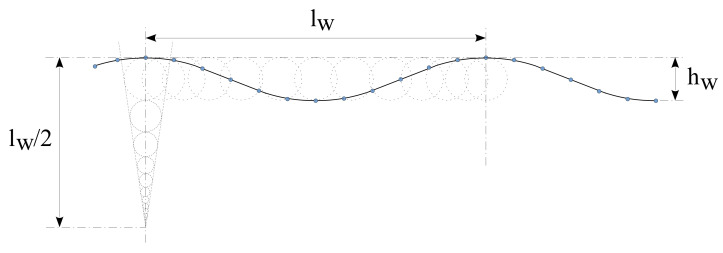
Ocean surface waves and deep current behavior.

**Figure 3 sensors-22-05513-f003:**
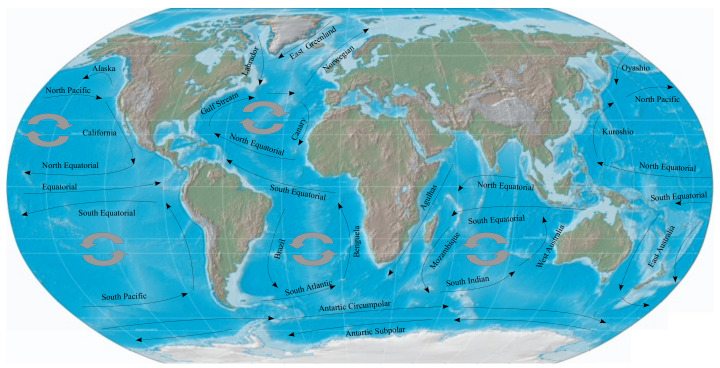
A map of the circulation of the world’s oceans.

**Figure 4 sensors-22-05513-f004:**
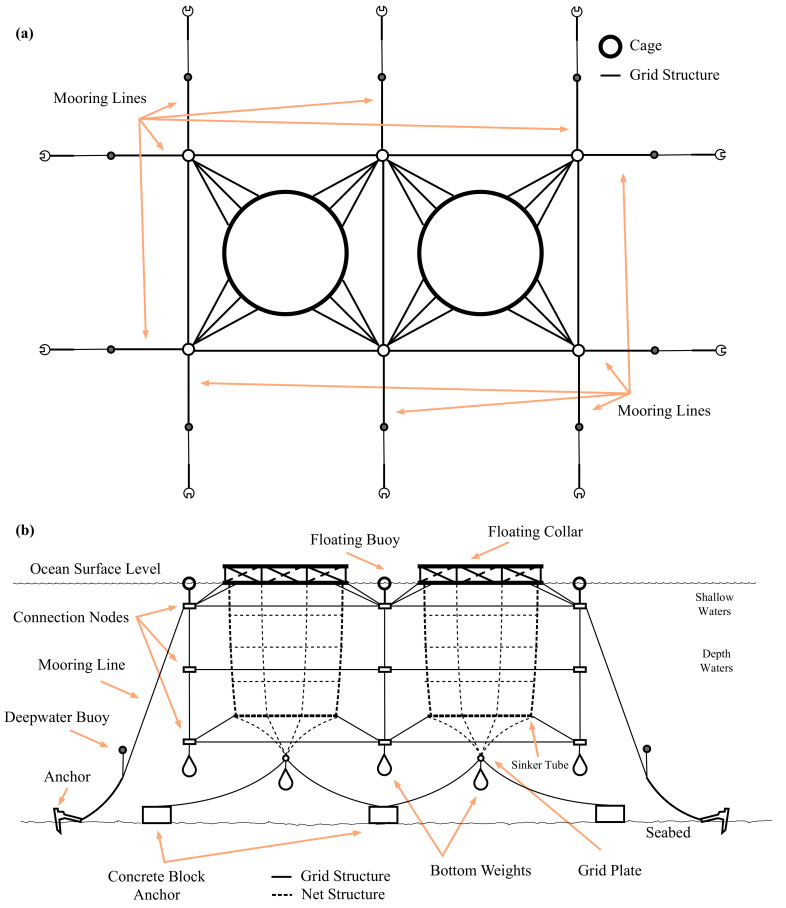
A 2 × 1 offshore aquaculture facility; (**a**) top view; (**b**) lateral view [26].

**Figure 5 sensors-22-05513-f005:**
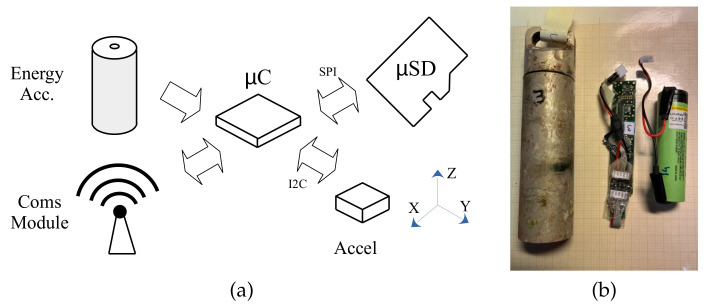
A block diagram of (**a**) the measurement unit architecture and (**b**) the fabricated prototype.

**Figure 7 sensors-22-05513-f007:**
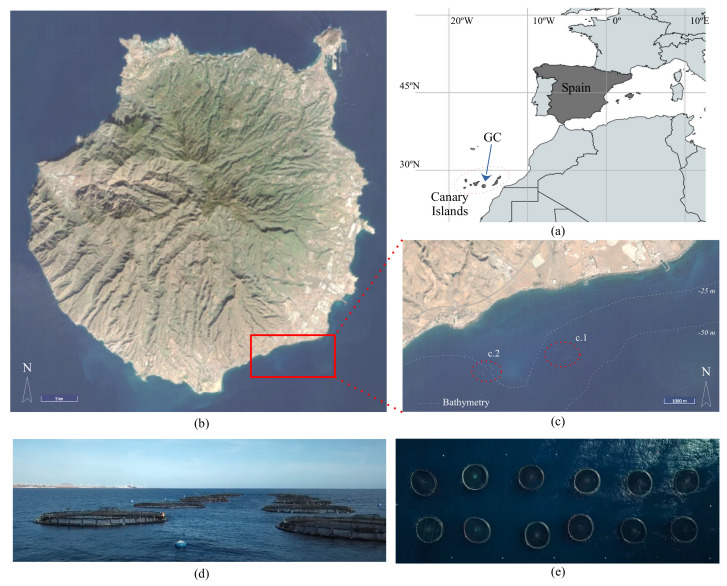
The study location that was used to evaluate the proposed MU: (**a**) a map of the Canary Islands in the Atlantic Ocean; (**b**,**c**) the satellite photos of Gran Canaria and the location of the offshore aquaculture facility in the south of GC ((c.1) the Aquanaria S.L. facility; (c.2) anther offshore aquaculture infrastructure), respectively; (**d**) a lateral view and (**e**) a top view of one of the 2 × 6 cage structure arrays that were used in our experiments.

**Figure 8 sensors-22-05513-f008:**
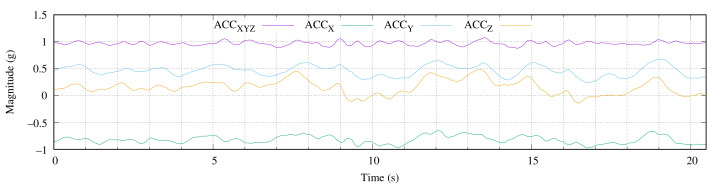
The raw data of the 256 acceleration measurements that were captured at a rate of 12.5 samples per second.

**Figure 9 sensors-22-05513-f009:**
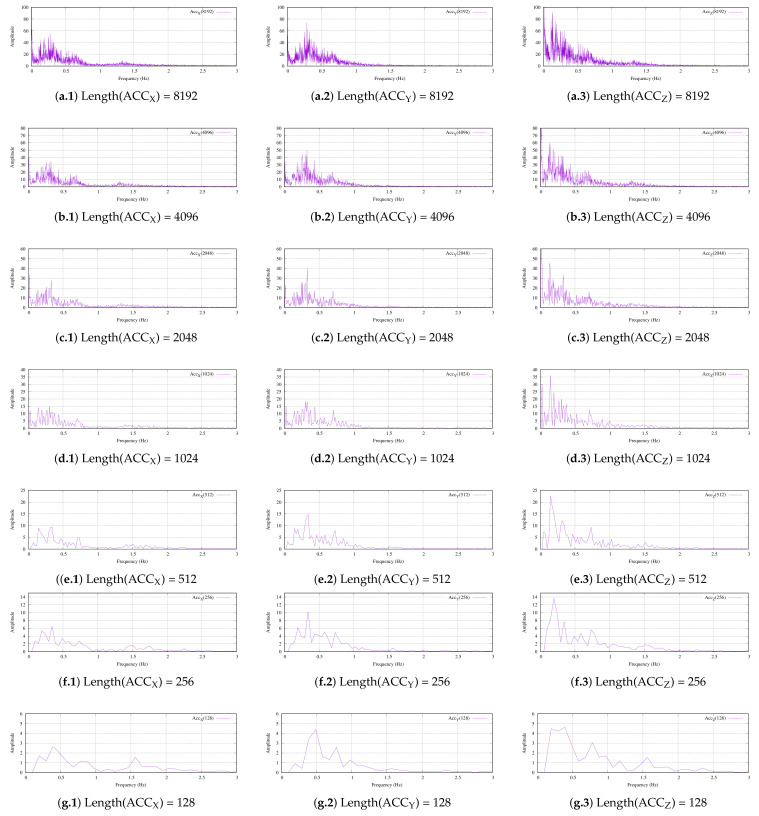
The frequency representations of the acquired data for the different acquisition lengths: (**a.i**) 8192 samples; (**b.i**) 4096 samples; (**c.i**) 2048 samples; (**d.i**) 1024 samples; (**e.i**) 512 samples; (**f.i**) 256 samples; (**g.i**) 128 samples. i is 1 to 3 for ACC_X_, ACC_Y_ and ACC_Z_, respectively.

**Figure 10 sensors-22-05513-f010:**
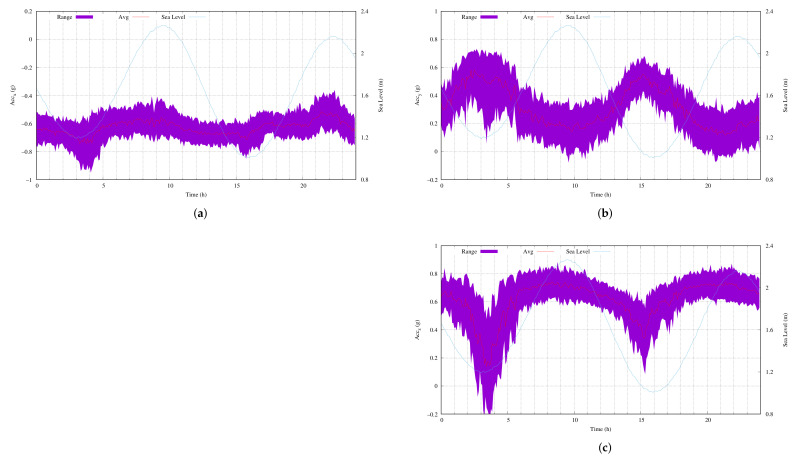
The acquired data for one day of acceleration samples: (**a**) ACC_X_; (**b**) ACC_Y_; (**c**) ACC_Z_.

**Figure 11 sensors-22-05513-f011:**
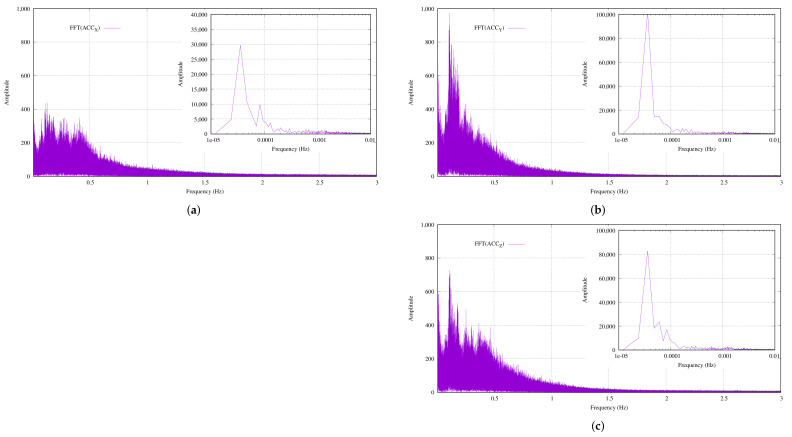
The frequency representations of the acquired data from the full day of measurements: (**a**) FFT(ACC_X_); (**b**) FFT(ACC_Y_); (**c**) FFT(ACC_Z_).

**Figure 12 sensors-22-05513-f012:**
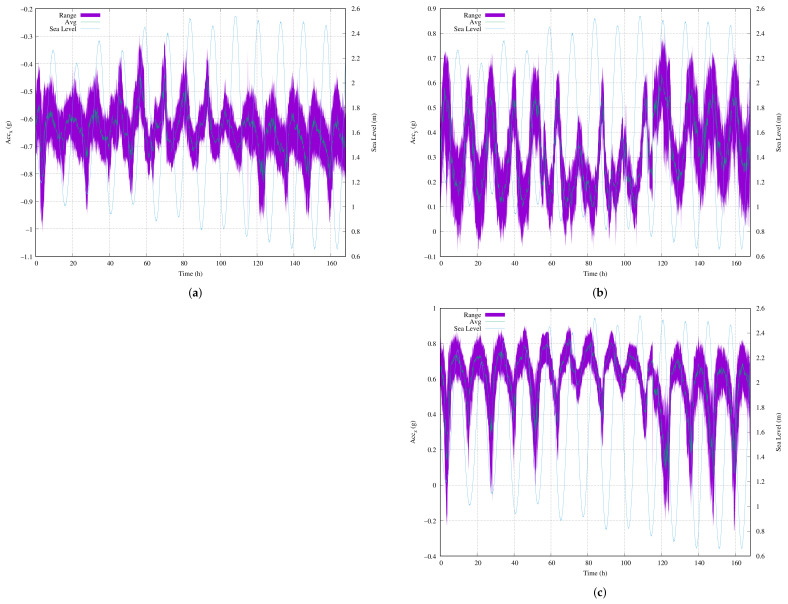
The time representations of the acquired data for the full week of measurements: (**a**) ACC_X_; (**b**) ACC_Y_; (**c**) ACC_Z_.

**Table 1 sensors-22-05513-t001:** The modes and power consumption of the selected devices.

Module	Mode	Current
CPU	VLPS	<1 μA
CPU	VLPR	<0.45 mA
ACC	Off	0
ACC	Stand-by	1.8 μA
ACC	Acquisition	6.0 μA
ACC	Data Transfer	0.57 mA
uSD	Sleep	<2.0 μA
uSD	Data Transfer	27 mA

**Table 2 sensors-22-05513-t002:** The power consumption of the worst-case scenario.

Module	Mode	Current	Time
CPU	VLPS	1 μA	325.48 s
CPU	VLPR	0.45 mA	2.2 s
ACC	Acquisition	6.0 μA	327.68 s
ACC	Data Transfer	0.57 mA	1.9 s
uSD	Sleep	2.0 μA	326.48 s
uSD	Data Transfer	0.57 mA	1.9 s
uSD	Data Writing	127 mA	1.9 s

**Table 3 sensors-22-05513-t003:** Comparisons between approaches in the literature, industry solutions and our approach.

Ref.	Method	Location	Year	Internal *F_s_* (Hz)	*F_s_* (Hz)	V*_max_* (m/s)	Accuracy (±% + cm/s)	Resolution (cm/s)	Size (mm × mm)	Weight (kg)	Data	Stand-Alone Runtime	Number of Samples	Cost (USD)
[32]	Doppler	Seabed	2022	23	4	5.0	1% + 1	2.5	75 × 500	2.3	μSD	4 Weeks	672	10–20 k
[33]	Doppler	Seabed	2022	250	64	7.0	1% + 0.1	2.5	75 × 824	3.2	μSD	6 Days	2304	10–20 k
[34]	Doppler	Seabed	2020	50	2	3.0	1% + 0.15	0.01	139 × 356	7.0	μSD	2 h	1	8–15 k
[35]	Turbine	Mooring	2020	-	(*1)	6.1	1%	NA	125 × 250	2.0	EC	NB	-	6–8 k
[36]	Tilt	Seabed	2015	8	1/60	0.8	3% + 3	0.1	27 × 750	0.34	μSD	4 Weeks	2.8	1.1–1.5 k
[37]	Tilt	Buoy	2022	1/120	1/120	1.1	NA	6.18	500 × 400	NA	IS	NB	-	2 k
[38]	Tilt	Mooring	2018	1	1	0.15	NA	NA	51 × 127	NA	μSD	56 Days	2.8	50
[39]	Tilt	Buoy	2014	1	1	0.6	4%	5.0	78 × 380	NA	IM	6.2 h	1.55	100
[14]	Tilt	Seabed	2020	10	1/60	1.0	20%	NA	245 × 2000	3.0	μSD	NB	-	NA
Ours	Acceleration	Mooring	2022	12.5	12.5	0.8 (*2)	0.85% + 0.035	0.01 (*3)	25.4 × 130	0.15	μSD	180 Days	13,500	50

*1, each turn produces a pulse; *2, assuming Δv = $\frac{1}{2}$Δa/*F_s_* and Δa = 2 g; *3, assuming Δv = $\frac{1}{2}$ Δa/*F_s_* and Δa = g/4096 = 0.2441 mg; NA, not available; NB, no battery mode; μSD, micro SD memory card; EC, external counter; IS, Internet server; IM, 21.4 K samples of internal memory.

## Data Availability

The data presented in this study are available on request from the corresponding author. The data are not publicly available due to industrial property confidentiality.

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
