# Peer review of "Novel Deep-Water Tidal Meter for Offshore Aquaculture Infrastructures"

_sensors, 2022, doi:10.3390/s22155513_

Round 1
Reviewer 1 Report
Please see the attached PDF. Thanks!

Reviewer 2 Report
In the present paper, the author presented a novel instrument based on inertial measurements for the Offshore Aquaculture site. It is a valuable manuscript. However, the present work was not organized in a clear structure and logical way. Thus, it is advised that the authors will be given the opportunity to perform major revisions of their paper. Before this paper can be properly considered for publication there are issues that must be dealt with:
1. Abstract section (Line 1- 13), The Abstract only presents the work done in the paper and should be improved. It should summarize not only the main work but also the interesting conclusions.
2. Introduction section (Line16-139), The logical flow of the author's statement is confusing and could be improved. Maybe the descriptions (Line 17-60) were unnecessary for this paper, given that the authors focused on measuring tidal currents around offshore aquaculture facilities. My suggestion is that the author first state the significance of measuring the current velocity around the aquaculture farm, then summarize the inadequacies of the current measuring instruments, and finally introduce the author's focus.
3. Line 110-112, The following papers can be added here to improve persuasiveness.
https://doi.org/10.1016/j.oceaneng.2015.04.045
https://doi.org/10.1016/j.oceaneng.2021.109941
4. Line 125-139, It can be better explained by citing some literature. The following papers could be referred to:
P. F. Lader and B. Enerhaug, "Experimental investigation of forces and geometry of a net cage in uniform flow," in IEEE Journal of Oceanic Engineering, vol. 30, no. 1, pp. 79-84, Jan. 2005, DOI: 10.1109/JOE.2004.841390.
https://doi.org/10.1016/j.aquaeng.2007.11.001
https://doi.org/10.1016/j.oceaneng.2021.108872
https://doi.org/10.1016/j.aquaeng.2013.09.006
5. Line 173, The author states” Figure 1, the angle q is function of the water speed where the device is submerged.” The reviewer really doesn’t understand this sentence in that part of the paper.
6. Fig. 3 seems needless here.
7. Line268-269: The author states” Finally, since the nets of the cages containing the fish act as sails, deep water currents must be taken into account.” The reviewer really doesn’t understand this sentence. What does this sentence mean? What's the causal relationship?
8. The advantages of the new measuring instrument proposed by the authors are not very clear.
9. Section 2.5, More information should be added here. For example,
What is the arrangement of the cages on this farm site?
Was there any farmed fish in cages during the measuring?
Where is the Sensor specifically set on the farm site?
10. Line544-552, Considering this is the conclusion section, it's better to be more concise.
11. In the Conclusion section, the author should demonstrate to the reader the utility of the instrument for offshore aquaculture Infrastructures.
Reviewer 3 Report
The authors propose an integrated ultra-low power system using an accelerometer to measure tidal currents in offshore aquaculture infrastructure from a practical point of view. It is a topic of interest in marine aquaculture for the improvement of the industry. It is desirable to strengthen the importance of knowing the currents for the control of environmental parameters, talking about the benefit of sustainable aquaculture, referring to the cost benefit of this development, and the reduction of the impact on the environment by knowing and scaling this technology. It is desirable to propose a section before the conclusions on these issues.
Round 2
Reviewer 1 Report
No comment.
Reviewer 2 Report
Thanks for your revising. I have checked the response from the authors and have no further comments.